

# Effect of apatinib on the pharmacokinetics of tramadol and O-desmethyltramadol in rats

Su-su Bao[1],*, Peng-fei Tang[1],*, Nan-yong Gao[2], Zhong-xiang Xiao[1], Jian-chang Qian[2], Long Zheng[1], Guo-xin Hu[2] and Huan-hai Xu[1]

[1] Affiliated Yueqing Hospital, Wenzhou Medical University, Wenzhou, China
[2] School of Pharmaceutical Sciences, Wenzhou Medical University, Wenzhou, China
* These authors contributed equally to this work.

## ABSTRACT

Since the combination of anticancer drugs and opioids is very common, apatinib and tramadol are likely to be used in combination clinically. This study evaluated the effects of apatinib on the pharmacokinetics of tramadol and its main metabolite O-desmethyltramadol in Sprague-Dawley (SD) rats and the inhibitory effects of apatinib on tramadol in rat liver microsomes (RLMs), human liver microsomes (HLMs) and recombinant human CYP2D6.1. The samples were determined by ultra-performance liquid chromatography-tandem mass spectrometry (UPLC-MS/MS). The *in vivo* results showed that compared with the control group, apatinib increased the $AUC_{(0-t)}$, $AUC_{(0-\infty)}$ and $C_{max}$ values of tramadol and O-desmethyltramadol, and decreased the values of $V_Z/F$ and $CLz/F$. In addition, the $MRT_{(0-t)}$, $MRT_{(0-\infty)}$ values of O-desmethyltramadol were increased. *In vitro*, apatinib inhibited the metabolism of tramadol by a mixed way with $IC_{50}$ of 1.927 μM in RLMs, 2.039 μM in HLMs and 15.32 μM in CYP2D6.1. In summary, according to our findings, apatinib has a strong *in vitro* inhibitory effect on tramadol, and apatinib can increase the analgesic effect of tramadol and O-desmethyltramadol in rats.

## INTRODUCTION

Tramadol is a centrally acting, fully synthetic opioid with an atypical mechanism of action, it is not only a μ-opioid receptor agonist, but also a serotonin and norepinephrine reuptake inhibitor (*Barbosa et al., 2016*). It was first synthesized in 1962 and it was made available to the foreign markets under the brand name Tramal for pain treatment in 1977 (*Grond & Sablotzki, 2004*). Until 1995, it was approved by the US Food and Drug Administration for the treatment of moderate to moderately severe pain in adults (*Hassamal et al., 2018*). In 2013, tramadol ranked second in total U.S. opioid market sales, accounting for 14.7% (*Miotto et al., 2017*). Typical side effects caused by opioids include gastrointestinal reactions (nausea, vomiting, constipation, *etc.*), itching, dizziness, hypogonadism, sleep disturbance, inattention, respiratory depression, *etc.* In addition to this, there are other effects related to tolerance, dependence, and addiction (*Burgess & Williams, 2010*; *Gunther*

Corresponding authors
Guo-xin Hu, hgx@wmu.edu.cn
Huan-hai Xu,
xhh19992000@163.com

*et al., 2018*; *Manchikanti et al., 2010*; *Ventura, Carvalho & Dinis-Oliveira, 2018*). Tramadol is generally considered to be a "weak opioid" (its agonistic effect on μ-opioid receptors is only one-fourth to one-tenth of that of morphine), which leads to an arguable perception of higher safety (*Hassamal et al., 2018*). However, due to the inhibition of serotonin and norepinephrine reuptake, tramadol has additional risks in addition to the side effects of opioids, including: epilepsy, tachycardia, serotonin syndrome, hypertension, and reports of mania (*Beakley, Kaye & Kaye, 2015*; *Reimann & Schneider, 1998*).

The degree of oral absorption of tramadol is almost 100%, after a single oral administration of tramadol 100 mg, the bioavailability is 68% because of the 20–30% first-pass metabolism. After multiple administrations (100 mg × 4), the bioavailability of tramadol increases to 90–100% (*Grond & Sablotzki, 2004*). The metabolism of tramadol is very complex. More than 80% of tramadol is metabolized by the liver (*Desmeules, 2000*). At present, 23 different metabolites (active and inactive) of tramadol have been identified in humans (*Grond & Sablotzki, 2004*; *Wu, McKown & Liao, 2002*). Among them, O-desmethyltramadol is the main active metabolite. The affinity of O-desmethyltramadol is 700 times higher than its parent compound tramadol and 5.5 times lower than morphine (*Gillen et al., 2000*). Tramadol is metabolized to O-desmethyltramadol by CYP2D6, and to N-desmethyltramadol by CYP 3A4 and 2B6. N-desmethyltramadol is pharmacologically inactive (*Gong et al., 2014*; *Grond & Sablotzki, 2004*). Therefore, changes in CYP450 will affect the metabolism of tramadol. Tramadol is excreted mainly through the kidneys (about 90%), about 30% of the dose is excreted in the urine as a prototype (*Grond & Sablotzki, 2004*; *Lassen, Damkier & Brosen, 2015*).

Apatinib (Aitan, brand name in China) is a small molecule tyrosine kinase inhibitor, independently developed by China and approved for the subsequent-line treatment of advanced gastric or gastroesophageal junction adenocarcinoma (*Hu et al., 2014*). As one of the latest oral anti-angiogenic drugs, apatinib has been applied to various types of malignant tumors, such as breast cancer, non-small cell carcinoma and epithelial ovarian cancer *etc.*, and has obvious survival benefits and tolerable toxicity (*Hu et al., 2014*; *Huang et al., 2016*; *Langer, Mok & Postmus, 2013*; *Lu et al., 2017*; *Miao et al., 2018*; *Zhang et al., 2021*). The bioavailability of oral apatinib ranges from 10% to 20% (*Geng & Li, 2015*). Apatinib is mainly metabolized by CYP3A4/5 and, to a lesser extent, CYP2D6, CYP2C9, and CYP2E1. After 96 h of oral administration of apatinib, 78% of the drug is excreted, 69.8% of the amount is found in feces and only 7.02% in urine (*Ding et al., 2013*). Cancer and pain are inseparable. Moderate to severe pain is common in cancer patients, affecting 70–80% of patients with advanced cancer (*Caraceni et al., 2012*). Opioids are still an essential tool for controlling cancer-related pain and are being promoted in international guidelines (*Edler-Buggy et al., 2020*). It seems that the combination of anticancer drugs and opioids is very common. Therefore, it is very likely that apatinib and tramadol can be combined clinically. In addition, a previous study found that apatinib can inhibit CYP2D6, CYP2C9, CYP3A4 and CYP2B6 (*Bao et al., 2018b*), and tramadol is mainly metabolized by CYP2D6. This suggests there may be an interaction between apatinib and tramadol.

In this study, *in vivo*, we determined the effect of apatinib on the pharmacokinetics of tramadol and O-desmethyltramadol in rats. *In vitro*, we identified the effect and

mechanism of apatinib on tramadol in human and rat liver microsomes (RLM and HLM) as well as recombinant human CYP2D6.1.

## MATERIALS AND METHODS

### Chemical and reagents

Tramadol (CAS:27203-92-5), O-desmethyltramadol (CAS:144830-15-9), apatinib (CAS: 811803-05-1) and midazolam (CAS:59467-70-8) (used as internal standard, IS) were bought from Shanghai Canspec Scientific & Technology Co., Ltd. LC–MS grade acetonitrile (ACN) and methanol were purchased from Merck (Darmstadt, Germany). Formic acid of HPLC grade (FA, purity 99.9%) was obtained from J&K scientific Ltd. (Shanghai, China). The reduced nicotinamide adenine dinucleotide phosphate (NADPH, coenzymes for *in vitro* incubation systems) was purchased from Roche Pharmaceutical Ltd. (Basel, Switzerland). Carboxymethylcellulose sodium salt (CMC-Na) was from Sigma-Aldrich Company (Shanghai, China). Pooled RLM and HLM (enzymes for simulating metabolism *in vitro*) were bought from Corning Life Sciences Co., Ltd. Recombinant human CYP2D6.1 and cytochrome b5 (coenzymes for *in vitro* incubation systems) were kind gifts from Beijing Hospital (Beijing, China).

### Equipment and operation conditions

Data were collected as previously described in *Bao et al. (2018a)*. Concentrations of tramadol and O-desmethyltramadol were determined by ultra-performance liquid chromatography-tandem mass spectrometry (UPLC-MS/MS). Mass spectrometer contained an Acquity UPLC XEVO TQD triple quadrupole (Waters Corp., Milford, MA, USA) and an electrospray ionization source. The chromatographic separation was performed on a Waters ACQUITY UPLC BEH C18 column ($2.1 \times 50$ mm, 1.7 μm; Waters Corp., Milford, MA, USA) at 40 °C. The transitions were $m/z$ 264.2 → 58.0 for tramadol, $m/z$ 250.2 → 58.2 for O-desmethyltramadol, $m/z$ 326.1 → 291.1 for IS. The mobile phase consisted of solvent A (0.1% formic acid) and solvent B (ACN) with gradient elution at a flow rate of 0.4 ml/min. After many attempts, we finalized a gradient elution program as follows: 0–1.4 min, 60–10% A, 1.4–2.6 min, 10–60% A. Under this procedure, the peak shape of midazolam, tramadol and O-desmethyltramadol can basically remain sharp and symmetrical, and the retention time is also very short, so that our final running time is only 2.6 min, saving time cost.

### *In vitro* experiments

The concentrations of the stock solutions of apatinib and tramadol (both diluted by methanol) were 2 and 4 mg/ml respectively. The 200 μl incubation system contained 1.6 μl apatinib, 3.0 μl tramadol, 5 μl rat liver microsomes (RLM, 0.70 mg/ml) or human liver microsomes (HLM, 0.31 mg/ml) or 5 μl CYP2D6.1 (1 pM) along with 3.7 μl b5 (5.25 μg/ml), and 10 μl NADPH (1 mM), and 180.4 μl 1M potassium phosphate buffer (PBS) for HLM and RLM, 176.7 μl PBS for CYP2D6.1. In the experiment of $IC_{50}$ determination, the concentration of apatinib was designed at 0.01, 0.1, 1, 10, 50 and 100 μM, while that of tramadol was 80 μM for RLM, 50 μM for HLM, 40 μM for

CYP2D6.1, which were close to their Km value correspondingly. In the experiment of the inhibitory effect of apatinib on tramadol, the concentration gradient of apatinib (0–32 μM) and tramadol (10–160 μM) was set according to the $IC_{50}$ value and the Km value. The incubation system was carried out at 37 °C, and after incubating for 30 min, it was cooled to −80 °C. Then 400 μl ACN and 20 μl IS (20 ng/ml) were added to the mixture. After vortex mixing for 2 min and high-speed centrifugation at 13,000 rpm for 10 min, the supernatant that maintained at cooled was taken for testing the concentration of tramadol.

### *In vivo* experiments

A total of 12 male Sprague–Dawley (SD) rats were purchased from the Shanghai Animal Experimental Center. All procedures conformed to the Guide for the Care and Use of Laboratory Animals and were conducted with the approval of the Animal Care and Use Committee of Wenzhou Medical University (wydw2018-0002). The animals were randomly divided into two groups ($n = 6$): group A was apatinib group (taking apatinib with tramadol), group B was the control group (taking tramadol alone). The SD rats were kept in the specific-pathogen-free (SPF) laboratory, and fed normally. Before the formal experiment, the 12 rats were fasted for 12 h but not banned from drinking water. At the beginning of the experiment, the group A was given 40 mg/kg apatinib orally (apatinib was dissolved in 0.5% CMC-Na), and the group B was given the same amount of 0.5% CMC-Na as the group A. After 30 min, both groups were given 20 mg/kg tramadol orally. In order to ensure the best sampling points, we conducted pre-tests before the formal test, and finally determined that the time points of blood collection from the tail vein of rats were 0.083, 0.25, 0.5, 1, 2, 4, 6, 8, 10, 12 and 24 h after administration of tramadol. After the end of the experiment, the rats were euthanized with anesthesia (Pentobarbital sodium 100–200 mg/kg), then they were packaged and finally incinerated per biological waste disposal protocol.

The above blood was centrifuged at 13,000 rpm for 10 min to obtain the required plasma. A total of 100 μl of the collected plasma was taken and mixed with 20 μl IS (300 ng/ml) and 200 μl ACN in a 1.5 ml centrifuge tube. After being vortexed for 2 min and centrifuged at 13,000 rpm for 10 min, the supernatant was diluted with pure water (1:1) for UPLC-MS/MS analysis.

### Statistical analysis

$IC_{50}$ and Lineweaver-Burk Plot are calculated by GraphPad Prism 5.0 (GraphPad, San Diego, CA, USA). The average concentration-time curve is drawn by Origin 8.0. The pharmacokinetic parameters were evaluated by DAS software (version 3.0), using non-compartmental analysis. The statistical analysis of all data is expressed as mean ± standard deviation and analyzed by SPSS 19.0. using independent samples t-test. $P < 0.05$ represents statistical significance.

## RESULTS

### UPLC-MS/MS

The correlation coefficients of the calibration curves of tramadol and O-desmethyltramadol were both greater than 0.99. Tramadol is used in a concentration range of 1 to 2,000 ng/ml, and O-desmethyltramadol is used in a concentration range of 0.25 to 500 ng/ml. The chromatograms of tramadol, O-desmethyltramadol and IS under different conditions are shown in Fig. 1. The methodological validation data are presented in the Tables 1–3.

### Effects of apatinib on the metabolism of tramadol *in vitro*

The $IC_{50}$ curve and the Lineweaver-Burk plot of apatinib on tramadol in HLM, RLM and CYP2D6.1 are shown in Figs. 2 and 3, respectively, and the corresponding values are shown in Table 4. According to Table 4, The $IC_{50}$ values of RLM and HLM are very close. The inhibitory strength of apatinib on tramadol is greater in RLM and HLM than in CYP2D6.1. The results indicate that apatinib inhibits tramadol in a mixed way.

### Effects of apatinib on the metabolism of tramadol *in vivo*

The average plasma concentration-time curves of tramadol and O-desmethyltramadol and their corresponding pharmacokinetic parameters are shown in Fig. 4, Tables 5 and 6. Apatinib inhibited the metabolism of tramadol and O-desmethyltramadol in rats and changed the pharmacokinetic parameters of tramadol and O-desmethyltramadol. Compared with group B (control group), the $AUC_{(0-t)}$, $AUC_{(0-\infty)}$ and $C_{max}$ values of tramadol and O-desmethyltramadol were increased in apatinib group, but Vz/F and CLz/F values of tramadol and O-desmethyltramadol were decreased. In addition, the $MRT_{(0-t)}$, $MRT_{(0-\infty)}$ values of O-desmethyltramadol were increased in apatinib group. Other parameters have no significant difference.

## DISCUSSION

Tramadol is mainly metabolized by CYP2D6, CYP3A4 and CYP2B6. Since the phase I metabolic reaction mediated by CYP450 is slower than the phase II binding reaction, they become the rate-limiter for the overall metabolic disposal of CYP substrate drugs (*Miotto et al., 2017*). CYP enzymes are easily induced and inhibited by other substrates, which means that the plasma concentration of tramadol and its main metabolite may be affected by drugs that affect the activities of CYP2D6, CYP3A4 and CYP2B6. Since apatinib is metabolized by CYP3A4/5, CYP2D6, CYP2C9 and CYP2E1 (*Ding et al., 2013*), sharing two CYP metabolic pathways with tramadol, and the fact that tramadol and apatinib are likely to be combined clinically, whether apatinib affects the metabolism of tramadol is worth studying.

The results of *in vivo* experiments show that apatinib does influence the metabolism of tramadol. Compared with the control group, the $AUC_{(0-t)}$, $AUC_{(0-\infty)}$ and $C_{max}$ of tramadol increased, Vz/F and CLz/F decreased, which proves that apatinib inhibits the metabolism of tramadol in rats, increasing the side effects of tramadol. The inhibitory effect of tramadol on the neuronal reuptake of norepinephrine and 5-HT increases its side

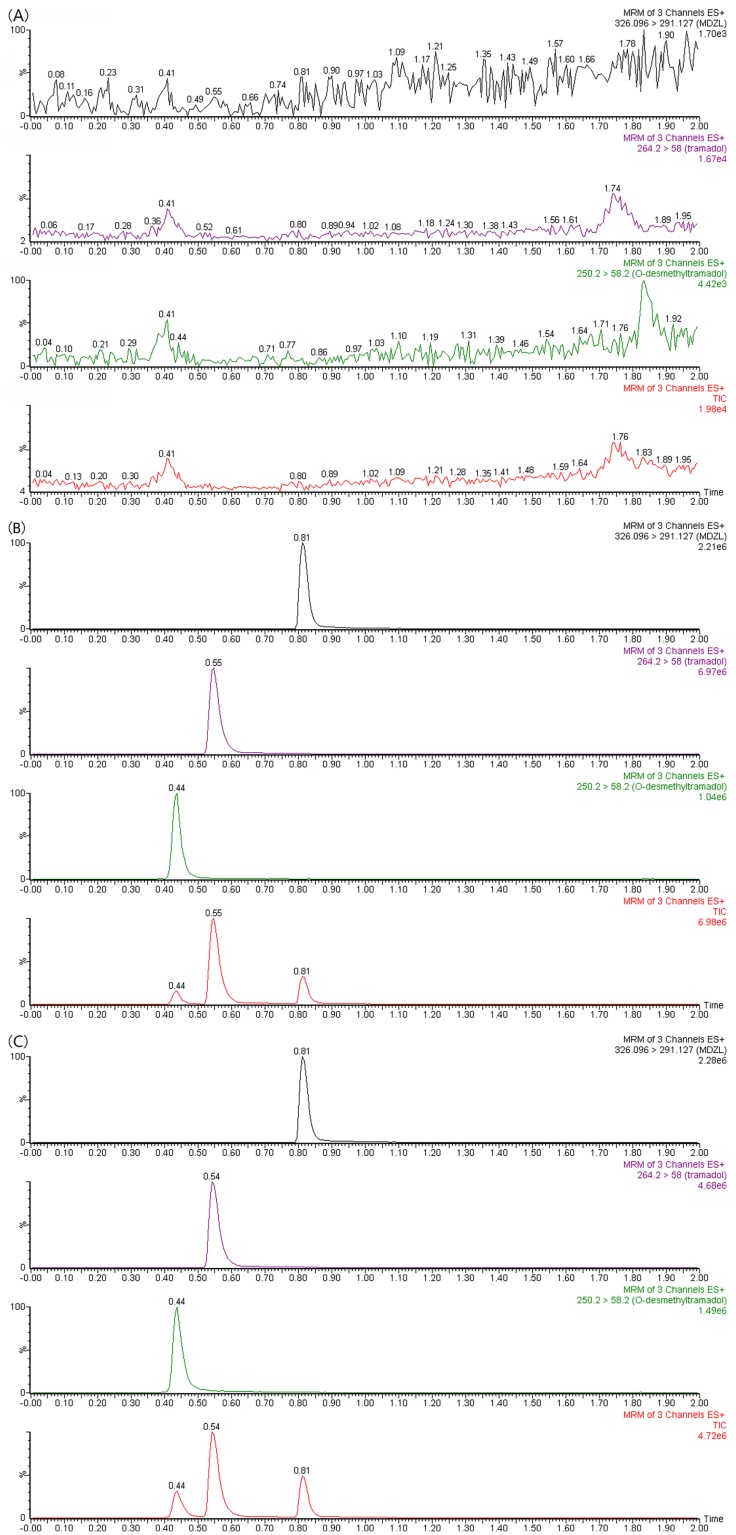

**Figure 1  UPLC-MS/MS chromatograms of tramadol, O-desmethyltramadol and IS (midazolam). (A)** Blank plasma sample. (B) Blank plasma spiked with 100 ng/mL tramadol, 25 ng/mL O-desmethyl­tramadol and 300 ng/mL IS. (C) Apatinib-treated rat plasma sample at 4 h after tramadol. TIC, total ionic chromatogram.           

Table 1 **Precision and accuracy for tramadol and O-desmethyltramadol of QC sample in rat plasma ($n = 6$).**

| Analytes | Level (ng/ml) | Intra-day precision | | Inter-day precision | |
|---|---|---|---|---|---|
| | | RE (%) | RSD (%) | RE (%) | RSD (%) |
| Tramadol | 1 | 8.75 | 8.89 | 9.52 | 3.45 |
| | 3 | 5.70 | 3.33 | 6.18 | 4.12 |
| | 150 | 8.42 | 2.98 | 6.39 | 4.44 |
| | 1,500 | 2.03 | 3.38 | 1.71 | 1.14 |
| O-desmethyltramadol | 0.25 | −9.28 | 5.57 | −6.52 | 3.01 |
| | 0.75 | 3.75 | 2.01 | −3.41 | 5.53 |
| | 50 | 9.43 | 2.05 | 8.34 | 5.60 |
| | 400 | 3.14 | 4.03 | 1.34 | 2.48 |

Table 2 **Recovery and matrix effect for tramadol and O-desmethyltramadol of QC sample in rat plasma ($n = 6$).**

| Analytes | Level (ng/ml) | Recovery (%) | Matrix effect (%) |
|---|---|---|---|
| Tramadol | 3 | 100.48 ± 6.89 | 98.06 ± 8.83 |
| | 150 | 102.65 ± 1.65 | 92.74 ± 2.02 |
| | 1,500 | 97.61 ± 7.30 | 104.20 ± 8.34 |
| O-desmethyltramadol | 0.75 | 96.93 ± 4.96 | 98.14 ± 5.70 |
| | 50 | 102.15 ± 7.17 | 106.31 ± 7.71 |
| | 400 | 102.81 ± 6.62 | 98.75 ± 5.20 |

Table 3 **Summary of stability of tramadol and O-desmethyltramadol in rat plasma under different storage conditions ($n = 6$).**

| Analytes | Level (ng/ml) | Room temperature | | 4 °C | | Three freeze-thaw | | −80 °C | |
|---|---|---|---|---|---|---|---|---|---|
| | | RE (%) | RSD (%) | RE (%) | RSD (%) | RE (%) | RSD (%) | RE (%) | RSD (%) |
| Tramadol | 3 | 4.98 | 5.03 | 2.05 | 7.95 | −3.25 | 9.76 | 5.17 | 7.74 |
| | 150 | 2.19 | 6.36 | 2.56 | 5.32 | −8.91 | 1.42 | 5.21 | 6.06 |
| | 1,500 | 6.12 | 8.44 | 6.39 | 7.56 | 3.27 | 9.64 | 4.85 | 7.73 |
| O-desmethyltramadol | 0.75 | −3.13 | 8.37 | −3.83 | 6.01 | −2.04 | 5.29 | 1.71 | 5.59 |
| | 50 | 5.31 | 7.55 | 3.48 | 6.51 | 6.04 | 7.38 | 5.86 | 10.09 |
| | 400 | 10.71 | 8.07 | 8.05 | 8.19 | 4.58 | 4.38 | 2.26 | 8.40 |

effect compared with other opioids (*Minami, Uezono & Ueta, 2007*). This *in vivo* result corresponds to the subsequent *in vitro* result. Apatinib strongly inhibited the metabolism of tramadol by a mixed way with $IC_{50} < 5$ μM in RLMs.

As we all know, because of the differences in species, animal experiments cannot be fully extrapolated to clinical applications, but they can provide some basic data for human research. In addition, although there are differences between species, there are similarities.

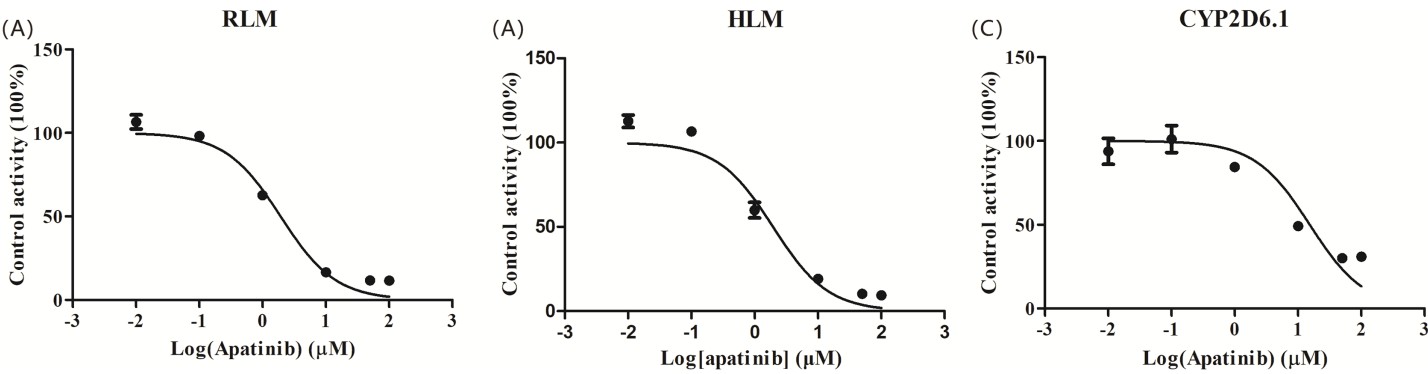

**Figure 2** Apatinib with various concentrations for half-maximal inhibitory concentration ($IC_{50}$) in the activity of (A) RLM, (B) HLM and (C) CYP2D6.1. Values are the mean ± SD, N = 3.

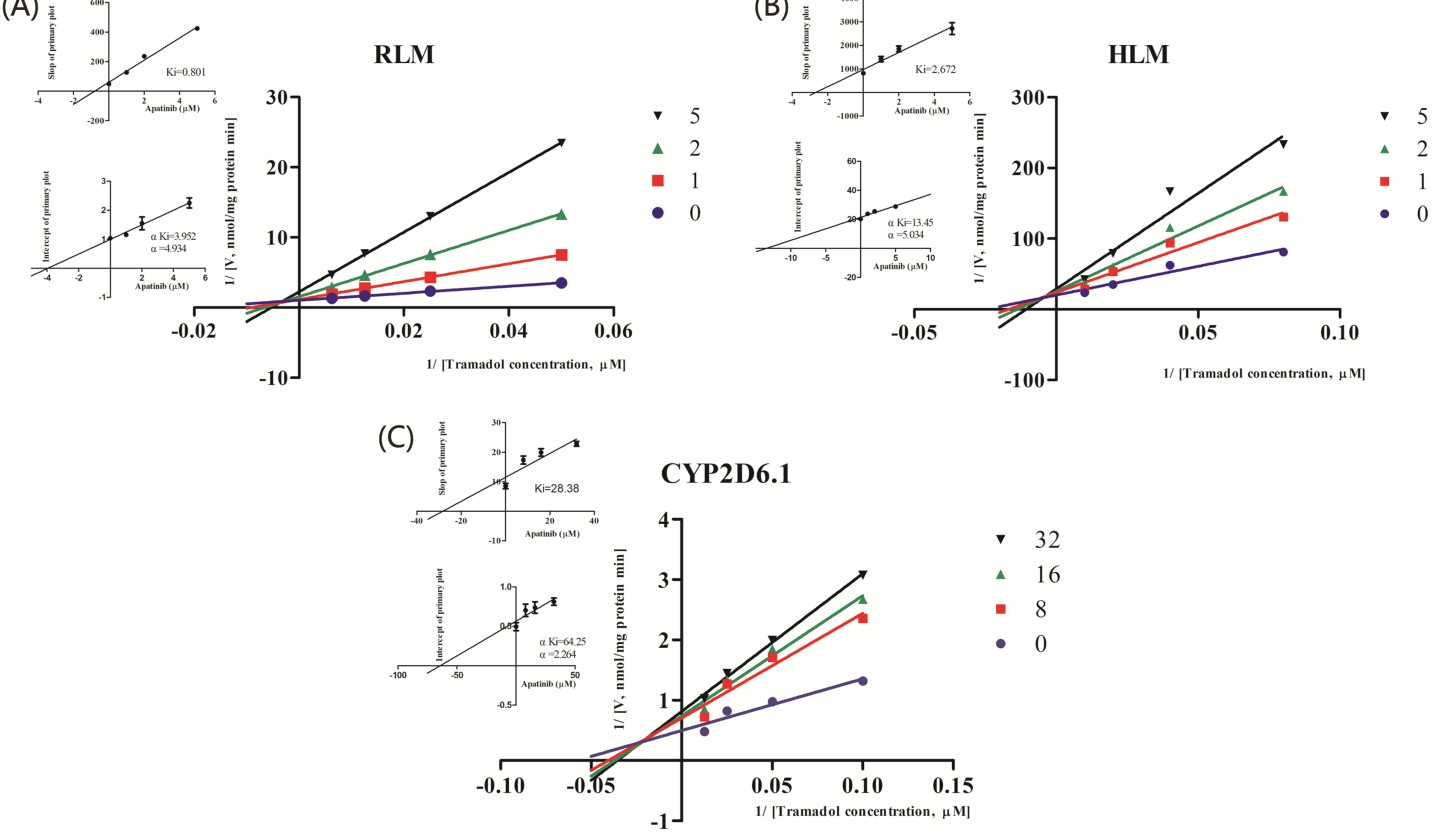

**Figure 3** Primary Lineweaver–Burk plot, the secondary plot for Ki and the secondary plot for αKi in the inhibition of the metabolism of tramadol by various concentrations of apatinib in (A) RLM, (B) HLM and (C) CYP2D6.1. Values are mean ± SD, N = 3.

Martignoni et al. concluded that CYP2E1 shows no large differences between species, and extrapolation between species appears to hold quite well, and the rat and human CYP2D isoforms share a high sequence identity (>70%) In our $IC_{50}$ experiment, the results showed that the values of $IC_{50}$ in HLMs (2.039 μM) are extremely close to those in RLMs (1.927 μM), which also increased the possibility that *in vivo* experiments in rats could be
**Table 4 The IC50 values and inhibitory effects of apatinib on tramadol in HLMs, RLMs and CYP2D6.1.**

|  | IC$_{50}$ values (μM) | Inhibition type | Ki (μM) | αKi (μM) | α |
|---|---|---|---|---|---|
| RLMs | 1.927 | Mixed | 0.801 | 3.952 | 4.934 |
| HLMs | 2.039 | Mixed | 2.672 | 13.45 | 5.034 |
| CYP2D6.1 | 15.32 | Mixed | 28.38 | 64.25 | 2.264 |

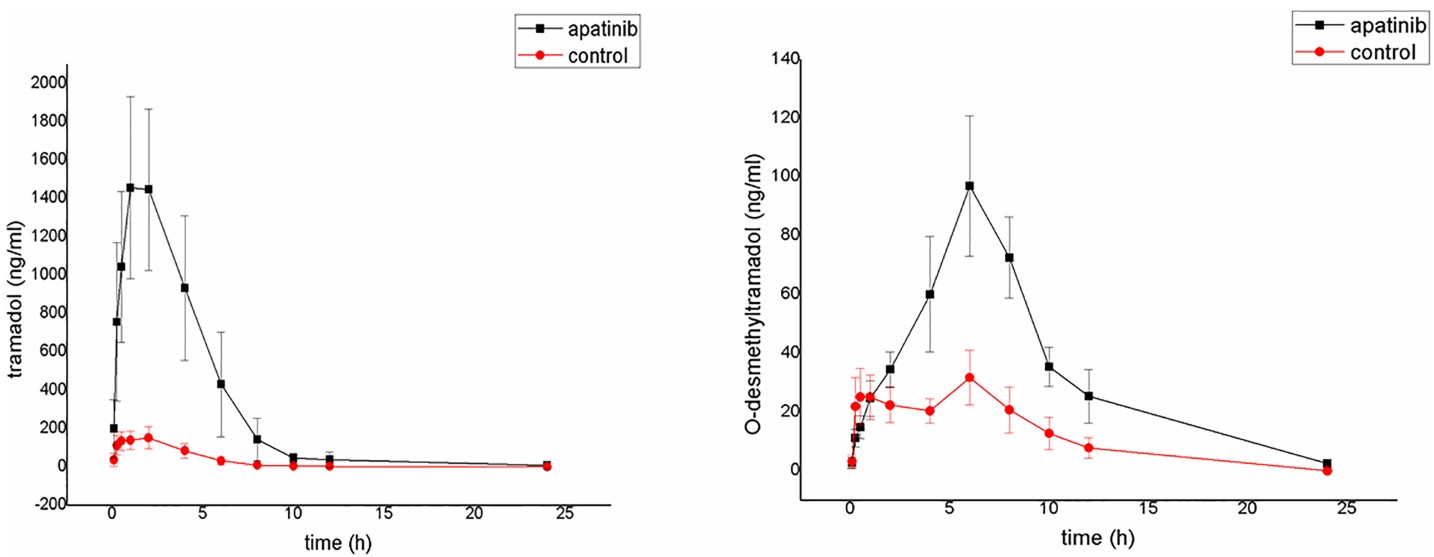

**Figure 4 Mean concentration-time curve of tramadol and O-desmethyltramadol in the control group and the apatinib group (N = 6).** Control group (tramadol alone) and the apatinib group (tramadol with apatinib) (N = 6).

**Table 5 The main pharmacokinetic parameters of tramadol in two groups ($n$ = 6).**

| Parameters | Group A | Group B |
|---|---|---|
| AUC(0–t) (ug/L * h) | 7,273.187 ± 2,680.883[**] | 710.148 ± 127.326 |
| AUC(0–∞) (ug/L * h) | 7,278.701 ± 2,683.291[**] | 724.840 ± 121.467 |
| MRT(0–t) (h) | 3.600 ± 0.377 | 3.389 ± 0.853 |
| MRT(0–∞) (h) | 3.617 ± 0.367 | 4.293 ± 1.619 |
| t1/2z (h) | 2.320 ± 1.072 | 7.554 ± 7.124 |
| Tmax (h) | 1.500 ± 0.548 | 1.792 ± 1.346 |
| Vz/F (L/kg) | 9.824 ± 4.610[*] | 324.580 ± 313.568 |
| CLz/F (L/h/kg) | 3.076 ± 1.151[**] | 28.327 ± 5.318 |
| Cmax (ug/L) | 1,555.169 ± 450.828[**] | 162.063 ± 50.829 |

Notes:
[*] $P < 0.05$ in comparison with group B.
[**] $P < 0.005$ in comparison with group B.
Group A (tramadol with apatinib), Group B (control group).

extrapolated to humans. A pharmacokinetic study of apatinib (after oral administration of 750 mg once daily) shows that the mean values of C$_{max}$ of apatinib is 3.8 uM (*Ding et al., 2013*), which is greater than its IC$_{50}$ value in HLMs. This indicates that at the usual clinical

**Table 6 The main pharmacokinetic parameters of O-desmethyltramadol in two groups ($n$ = 6).**

| Parameters | Group A | Group B |
|---|---|---|
| AUC(0–t) (ug/L * h) | 755.152 ± 208.119** | 246.079 ± 56.316 |
| AUC(0–∞) (ug/L * h) | 790.146 ± 187.052** | 281.735 ± 72.841 |
| MRT(0–t) (h) | 7.172 ± 0.693** | 5.289 ± 0.461 |
| MRT(0–∞) (h) | 7.776 ± 0.409* | 6.603 ± 0.896 |
| t1/2z (h) | 3.159 ± 0.768 | 2.925 ± 0.593 |
| Tmax (h) | 6.333 ± 0.816 | 4.125 ± 2.906 |
| Vz/F (L/kg) | 117.125 ± 23.395** | 309.179 ± 58.566 |
| CLz/F (L/h/kg) | 26.373 ± 5.473** | 77.049 ± 28.893 |
| Cmax (ug/L) | 97.216 ± 23.528** | 34.456 ± 8.107 |

Notes:
* $P < 0.05$ in comparison with group B.
** $P < 0.005$ in comparison with group B.
Group A (tramadol with apatinib), Group B (control group).

dose, the plasma concentration of apatinib in humans is sufficient to cause its inhibition of the metabolism of tramadol. Studies have pointed out that tramadol is metabolized much faster in animals than in humans: 1% and 25–30% of the oral dose are excreted in urine as prototypes respectively (*Grond & Sablotzki, 2004*; *Lintz et al., 1981*). This also implies that when apatinib and tramadol are used in combination in humans, the accumulation of tramadol in humans is more serious than in rats.

Although CYP2D6 only accounts for ∼2–4% of all liver CYP enzymes, it metabolizes about 25% of clinically used drugs, and about 80% of tramadol is metabolized by CYP2D6 (*Jin et al., 2013*; *Miotto et al., 2017*; *Zhou, 2009a*, *2009b*). Based on this, we studied the potential inhibitory effects of apatinib on tramadol in CYP2D6.1. Apatinib inhibits tramadol by a mixed way in CYP2D6.1, suggesting the complexity of the inhibition way. The inhibitory intensity of apatinib in 2D6.1 is much smaller than that of RLM and HLM, the reason may be that the inhibition of tramadol by apatinib *in vitro* is not only through CYP2D6. Studies have shown that apatinib can strongly inhibit CYP3A4 and CYP2B6 (*Bao et al., 2018b*) while the N-demethylation process of tramadol is through these two enzymes (*Kirchheiner et al., 2008*). Therefore, apatinib may also inhibit the metabolism of tramadol through CYP3A4 and CYP2B6.

In addition, compared with the control group, the exposure of O-desmethyltramadol to AUC$_{(0–t)}$, AUC$_{(0–∞)}$, MRT$_{(0–t)}$, MRT$_{(0–∞)}$ and C$_{max}$ was increased significantly, This may be because apatinib significantly reduced tramadol's first pass elimination, which greatly increased the amount of absorption. Besides, O-desmethyltramadol will subsequently be metabolized into N,O-didesmethyl-tramadol through CYP3A4 and CYP2B6 (*Subrahmanyam et al., 2001*), and this process may be hindered by apatinib (*Bao et al., 2018b*). Since O-desmethyltramadol is the main active metabolite, and its affinity is 700 times higher than that of its parent compound, exaggerated effects or even opioid intoxication may occur.

Patients often take more than one drug at a time. Accidental, unrecognized, or poorly managed drug-drug interaction (DDI) is a significant cause of morbidity and mortality

associated with prescription drug use (*Food and Drug Administration, 2017*). We chose apatinib and tramadol as the study subjects because these two drugs are likely to be taken together by patients with cancer pain. Useful information on potential DDI can be provided by studying the effects of apatinib on the metabolism of tramadol and O-desmethyltramadol. In addition, our results of *in vitro* experiments, along with pharmacokinetic data, provide mechanistic information that can inform the necessity and proper design of potential future clinical studies (*Food and Drug Administration, 2020*).

## CONCLUSIONS

In conclusion, based on our research results, apatinib can enhance the analgesic effect of O-desmethyltramadol, but also greatly increase the toxic and side effects caused by the accumulation of tramadol and O-desmethyltramadol. In order to avoid the risk of greatly increased adverse reactions, we do not recommend simultaneous administration of apatinib and tramadol clinically. Due to the high probability of combined application of apatinib and tramadol in cancer patients, our study provides a contribution to the rational use of drugs in this regard.

## ACKNOWLEDGEMENTS

Su-su Bao would like to thank Zhen Xu for his care and support in the process of writing and living.

### Funding

This work was supported by the Wenzhou Municipal Science and Technology Bureau (Y2022089), the National Key Research and Development Program of China (2020YFC2008301), the National Natural Science Foundation of China (NSFC 81973397), the Natural Science Foundation of Zhejiang Province (LBY21H070001) and the Wenzhou Municipal Science and Technology Bureau (Y20180594). The funders had no role in study design, data collection and analysis, decision to publish, or preparation of the manuscript.

### Grant Disclosures

The following grant information was disclosed by the authors:
Wenzhou Municipal Science and Technology Bureau: Y2022089.
National Key Research and Development Program of China: 2020YFC2008301.
National Natural Science Foundation of China: NSFC 81973397.
Natural Science Foundation of Zhejiang Province: LBY21H070001.
Wenzhou Municipal Science and Technology Bureau: Y20180594.

### Competing Interests

The authors declare that they have no competing interests.

## Author Contributions

- Su-su Bao conceived and designed the experiments, analyzed the data, authored or reviewed drafts of the article, and approved the final draft.
- Peng-fei Tang conceived and designed the experiments, prepared figures and/or tables, and approved the final draft.
- Nan-yong Gao performed the experiments, prepared figures and/or tables, and approved the final draft.
- Zhong-xiang Xiao performed the experiments, prepared figures and/or tables, and approved the final draft.
- Jian-chang Qian performed the experiments, prepared figures and/or tables, and approved the final draft.
- Long Zheng analyzed the data, authored or reviewed drafts of the article, and approved the final draft.
- Guo-xin Hu conceived and designed the experiments, authored or reviewed drafts of the article, and approved the final draft.
- Huan-hai Xu conceived and designed the experiments, authored or reviewed drafts of the article, and approved the final draft.

## Animal Ethics

The following information was supplied relating to ethical approvals (*i.e.*, approving body and any reference numbers):

All procedures conformed to the Guide for the Care and Use of Laboratory Animals and were conducted with the approval of the Animal Care and Use Committee of Wenzhou Medical University (wydw2018-0002).

## Data Availability

The raw measurements are available in the Supplemental File.

## Supplemental Information

Supplemental information for this article can be found online at http://dx.doi.org/10.7717/peerj.16051#supplemental-information.

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
