# Peer review of "Effect of apatinib on the pharmacokinetics of tramadol and O-desmethyltramadol in rats"

_PeerJ, doi:10.7717/peerj.16051_

## Round 0.1 · original submission · Major Revisions

Besides addressing the issues mentioned by our reviewers (#2 and #4 are particularly important) you should take note of the following issues I observed:

A) You used midazolam as an internal standard. How can you be sure that its concentration did remain constant, and that it was not metabolized in the time-course of the experiment?

B) in section 2.3, amounts of apatinib, tramadol, CYP2D6.1 and rat liver microsomes are in microliter, but the concentrations of the respective stock solutions are not given. Authors here state "the supernatant was taken for testing", but they do not state whether the activity graphs refer to "tramadol used" or "O-desmethyltramadol" produced. This must be clarified: for example, it appears from figure 4 that in the absence of apatinib hardly any O-desmethyltramadol is formed, which suggests that the metabolism they are observing is NOT due to CYP2D6.1, and therefore we need to know exactly what reaction(s) the authors are following in Figs 2/3 to ascertain whether a comparison of the inhibition parameters with CYP2D6.1 makes sense (or not)

C) The title of table 3 mentions olmutinib, although the paper itself used apatinib instead. Is this a typo or have the authors placed here, by mistake, a table belonging to a different study?

Reviewer 1 ·

Basic reporting

-Clear and unambiguous, professional English used throughout
-Literature references, sufficient field background/context provided
-Professional article structure, figures, tables

Experimental design

Research question well defined, relevant & meaningful. It is stated how research fills an identified knowledge gap

Validity of the findings

All underlying data have been provided; they are robust, statistically sound, & controlled

Annotated reviews are not available for download in order to protect the identity of reviewers who chose to remain anonymous.

Reviewer 2 ·

Basic reporting

The authors have studied the effect of apatinib on tramadol metabolism.

Experimental design

The authors have studied the effect of apatinib on tramadol metabolism.

Major comments:
Why choose apatinib as the inhibitor?

Which drugs had been studied as the inhibitors in the current study.

The IC50 value for apatinib to inhibit tramadol metabolism was about 7 times higher in CYP2D6.1 compared to that in human microsomes, which is higher than 10. So, this inhibition is weak inhibition, then it's not clinically significant.

Tramadol is mainly metabolized by CYP2D6, CYP3A4 and CYP2B6. So, added the data of apatinib on tramadol metabolism in CYP3A4 and CYP2B6.

There have been concerns in the literature regarding topic stating that there are differences between rat and human CYPs (e.g. Martignoni et al Expert Opin. Drug Metab. Toxicol. 2006;6:875-894).


The methodological validation data is worrying. Tramadol is used in a concentration range of 0.25 to 500 ng/ml, and O-desmethyltramadol is used in a concentration range of 1 to 1000 ng/ml. But, the peak of tramadol is about 2000 ng/ml, most are out of the range. Also, for the QC, do not meet FDA requirements.

What about long-term stability?

The English language needs correction and improvement.

Validity of the findings

The authors have studied the effect of apatinib on tramadol metabolism.

·

Basic reporting

Good

Experimental design

Good

Validity of the findings

Good

Additional comments

None

Reviewer 4 ·

Basic reporting

This is an interesting manuscript containing findings about the potential drug-drug interaction of apatinib and tramadol, which have been approved in several countries. A scientifically valid and generally accepted methodology was used to evaluate the relevant content, and results that could be helpful to support clinical use. If the following points are added, the reader's understanding will be further improved.

Experimental design

I suggest to provide the rationale for this desgin, like materials, sample size, and Line 92 and Line 118.

Validity of the findings

The conclusion drawn from this study cannot be easily extrapolated to clinical use. Please elaborate this part..

Additional comments

1. Your introduction needs more detail. I would recommend you to provide more clinical/preclinical pharmacokinetic data of tramadol at least including the absorption rate, metabolism fraction of CYP450, elimination pathway. Same for apatinib.
2. Please provide more justification for your study.

---

## Round 0.2 · Minor Revisions

First, on behalf of PeerJ I would like to apologize for this halfway change of scientific editor.

I've considered the manuscript and all the numerous change requests. I generally find the answers given to detail questions to be credible, and can understand the innovation height of this manuscript. I have chosen to ignore criticism based solely on the grounds that the study is based on an animal model, since it very clearly is indicated as such, and - despite some translation troubles - much valuable research and insights have been made from animal models.

I have chosen a "minor revision" call here, to give the authors a chance to go through some of the remaining (minor!) details, e.g. double check if units of concentrations are correct and double-check that there are no typos in such important places as the names of compounds. However, if these (minor) double-checks are made I believe this manuscript is acceptable for publication.

Reviewer 2 ·

Basic reporting

The validation data of this method is unconvincing, and the author's answer is not credible. In addition, the authors did not add other data about apatinib on tramadol metabolism for CYP3A4 and CYP2B6. Therefore, we think it was not to be accepted for this article.

Experimental design

see above

Validity of the findings

see above

Reviewer 4 ·

Basic reporting

The revised manuscript looks good.

Experimental design

All my comments on design have been addressed well.
No further comment from my end.

Validity of the findings

All my comments on the findings and discussion part have been addressed well.
No further comment from my end.

Additional comments

No further.

Reviewer 5 ·

Basic reporting

Experimental design

The experimental design seems sound and generally follows accepted methodology in the field. However, certain aspects of the design require further explanation and rationale.
Major Concerns:
• There is a lack of rationale provided for the selection of materials, sample size, and specific lines of the study design. A clear justification for these experimental choices is critical for the reader to assess the validity of the study. The authors may want to add the necessary rationale and justifications for these choices in the experimental design section.
• Some key experimental details are missing. For example, the exact number of replicates performed for each experiment should be specified to allow for assessment of statistical power. The authors need to clarify the number of replicates performed in each experiment.
Minor Concerns:
• It would be beneficial to provide more detail on how the sampling time points were determined for obtaining the complete plasma concentration-time curve. The authors are recommended to elaborate on the method and rationale behind the selection of sampling time points.

Validity of the findings

The findings of the study are interesting and appear valid based on the methodology used. Nevertheless, more robust explanation and discussion of their clinical implications and limitations are needed.
Major Concerns:
• The authors state that the results from the rat experiments can be extrapolated to humans to some extent, however, a more comprehensive discussion of the potential limitations and challenges of this extrapolation is necessary. The authors may want to expand the discussion section to include potential limitations of interspecies extrapolation.
• The IC50 values between human and rat samples are close, but it would be useful to discuss the variability and potential significance of the difference. The authors are recommended to discuss the variability in IC50 values and the potential implications of this in the results section.
Minor Concerns:
• It would be beneficial for the authors to clarify how they have controlled for potential confounding variables in their experiments. The authors may want to describe any steps taken to control for potential confounders in the methodology section.
• The authors should address potential limitations of their study, such as the possibility of other pathways contributing to the drug-drug interaction that have not been investigated. The authors may want to discuss these potential limitations in the discussion section.

Additional comments

This paper provides valuable insight into the interaction between apatinib and tramadol, however, there are areas for improvement such as emphasizing the clinical importance of the findings and improving the presentation of the manuscript. While the paper does provide insight into the potential interaction between apatinib and tramadol, it does not sufficiently emphasize the importance and relevance of these findings for clinical practice. The authors should elaborate on this aspect, particularly considering the therapeutic importance of both drugs. The authors are recommended to emphasize the clinical importance of these findings in the discussion and conclusion sections.

In summary, this study provides interesting and potentially valuable data on the interaction between apatinib and tramadol. However, it requires major revisions to improve the clarity of the text, enhance the experimental design and methodology sections, and to provide a more nuanced discussion of the results.

Annotated reviews are not available for download in order to protect the identity of reviewers who chose to remain anonymous.

·

Basic reporting

Thre are some gaps found in writing and language, mentioned in the Additional comments section below.

Experimental design

Most of the experiments are designed appropriately, however, there are couple of gaps, mentioned in the below additional comments section.

Validity of the findings

The findings are very useful while performing the clinical studies for Apatinib or Apatinib-Tradamol combination.

Additional comments

1. Appropriate references are required for these lines, lines 39-40, 49-50, 70-71.
2. Line 53, it should be “(Desmeules 2000). At present,” not be “(Desmeules 2000).At present,” (space)
3. The experiments for group A, group B were carried out with 6 SD rats each, however, from figure 4, the error bars seem pretty much huge, especially for apatinib group. Is the number of objects is sufficient? Should be discussed thoroughly.
4. Line 135, the administered dosage for apanitib is 40 mg/kg. Should provide the reasoning for choosing this specific dosage?
5. Line 136, for the group B, for the 0.5% CMC-Na, it is a negative control/placebo, but not “dosage”. Hence should be corrected.
6. For the 20 mg/kg dose of tramadol, reasoning should be provided.
7. Lines 163-164, its confusing to reader, should we check Table 1 or Table 4? (According to Table 1, The IC50 values of RLM and HLM are very close (Table 4) ).
8. Line 201, there are two full stops, should be corrected.
9. Table 5 and Table 6, the significant digits in the error bar is very significantly smaller than the actual error, they should be corrected based on error. (Example, table 5, Group A AUC (0-t) 7273.187±2680.883, the error is 2680.883, it can be just 2680 as the error is very high)
10. Table 4, it should be IC50 not IC50, should be corrected.
11. The reason for much lower α in CYP2D6 in table 4, should be discussed in the main text.
12. From figure 1, the retention time difference between tradamol and IS is very less. Was every sample confirmed by MS analysis or just the TIC profile?
13. Figure 1B has four figures however, the description was only provided for three (tramadol, O-desmethyltradamol and IS). Should provide information about the fourth chromatogram.
14. Line 251 should be corrected. It should be “2022).” not “2022, ).”
15. Why in vivo studies were not conducted for apanitib alone? Should be justified.

·

Basic reporting

I. Line 113, what was the solvent used to make apatinib and tramadol stock solutions?
II. Line 124, was the centrifugation of invitro incubation mixture maintained at room temperature or at cooled to collect supernatant for LC-MS-MS analysis?

Experimental design

I. 2.2. Equipment and operation conditions: for the LC-MS-MS determination of tramadol why midazolam chosen as internal standard over tramadol-d6?
II. 2.3 . In Vivo Experiments: can the accumulated dose of tramadol and O-desmethyltramadol measured to assess toxicity? May need explanation

Validity of the findings

I. This manuscript examined Effect of Apatinib on the Pharmacokinetics of Tramadol and Odesmethyltramadol in Rats that would help in assessing clinical applications.
II. LC-MS-MS concentration data for tramadol and O-desmethyltramadol determined to find Pharmacokinetic parameters supports the hypothesis.
III. Conclusions were drawn for the data that was summarized.

---

## Round 0.3 · accepted · Accept

The previous Academic Editor is no longer available so I went through the manuscript and the comments. Not all comments have been addressed, but to the best of my knowledge, those that have not been addressed are arguably outside of a reasonable expectation for a paper of this nature. I am also convinced that the authors have done a commendable job in double-checking and ensuring all values and descriptions are accurate.